# NTK-LoRA: Calibrating Fine-tuned Vision Transformers using Gaussian Processes

## Abstract

Fine-tuning remains essential for adapting foundation models to domains where high precision is required, such as medical imaging or autonomous driving. However, this often leads to overconfident and poorly calibrated models, especially when fine-tuned on small datasets. We propose NTK-LoRA, a simple and effective post-hoc calibration method for fine-tuned Transformer models (e.g., Vision Transformers and LLMs) that leverages the Gaussian process view of neural networks to perform Laplace approximation of the posterior. Our method is almost as straightforward to implement as temperature scaling (TS), requires no hyperparameter tuning or deeper expertise, allows incorporating prior knowledge through the choice of GP kernel, achieves better or comparable performance to TS and consistently outperforms Laplace calibration, which in our experiments often fails to improve over the baseline on binary classification.

## 1 Introduction

Deep learning has advanced rapidly in the last years, leading to powerful foundational models in both vision and language. Vision Transformers (ViT) (Dosovitskiy et al., 2021) and their successors have become standard backbones in computer vision, while models like GPT-4 (Achiam et al., 2023) and LLaMA-3 (Grattafiori et al., 2024) exemplify progress in large language models (LLMs). Pretrained on massive datasets, these models show strong zero- and few-shot performance, yet fine-tuning remains essential for high-precision, domain-specific applications.

Fine-tuned models often suffer from overconfidence, even when their predictions are incorrect. During fine-tuning, the model's parameters become overly specialized to the small dataset, leading to overly confident outputs for inputs it has not encountered before (Hendrycks & Gimpel, 2017; Guo et al., 2017). Correct calibration is particularly important when predictions are used to make decisions in critical applications such as healthcare, finance, engineering, or autonomous systems. This overconfidence may undermine their reliability.

Various methods have been proposed to address overconfidence in deep learning. Temperature scaling Guo et al. (2017) is a widely used post-hoc calibration method due to its simplicity and effectiveness, but it is limited to scaling logits and cannot capture more complex forms of miscalibration. Bayesian approaches treat model weights as random variables and estimate their posterior distribution, which can lead to improved calibration; however, exact inference is computationally intractable even for small networks and requires approximations.

A classical approach, originally proposed for small models, is the Laplace approximation (MacKay, 1992), which approximates the posterior distribution of neural network weights with a Gaussian. While effective for smaller networks, the Laplace approximation becomes less effective for modern Transformer-based architectures, as (i) covariance factorization is required, which can affect performance, and (ii) simple priors such as isotropic Gaussians can lead to pathological behavior, partly due to the high dimensionality of the parameter space (Cinquin et al., 2021).

Recent work Yang et al. (2023) has applied the Laplace approximation to fine-tuned large language models (LLMs) such as LLaMA-2 Touvron et al. (2023) using Low-rank Adaptation (LoRA) Hu et al. (2021). LoRA enables efficient adaptation of pretrained Transformer-based models by training

---

**Algorithm 1** High-level overview of our method

---

**Input**:
- Model: Transformer fine-tuned using LoRA for binary classification (e.q. ViT or LLM)
- Calibrated weights: $w$ (all trainable weights during finetuning or subset)
- Data: Training inputs $X_{train}$, binary labels $Y_{train}$ and test inputs $X_{test}$

**Algorithm**:
    **for all** inputs $x \in X_{train} \cup X_{test}$ **do**
        Compute Jacobian $J_w(x)$ of output logits w.r.t. weights $w$
        Reshape $J_w(x)$ into feature vector $\tilde{\phi}(x)$
    **end for**
    Fit a GP on normalized feature vectors $\{\phi(x) : x \in X_{train}\}$ and labels $Y_{train}$
    Compute calibrated probabilities and uncertainty for $X_{test}$

---

only a small number of parameters, to which Laplace can then be applied. However, even with LoRA, covariance factorizations (e.g., diagonal or Kronecker) are still required for tractability Ritter et al. (2018); Daxberger et al. (2021), and implementing these efficiently for large models remains challenging in practice.[1]

Applying the Laplace approximation has been shown to be equivalent to Gaussian process (GP) inference with the Neural Tangent Kernel (NTK) that plays an important role in the theoretical understanding of neural networks Lee et al. (2018); Garriga-Alonso et al. (2019); Novak et al. (2019). Building on this insight, recent work has proposed efficient variants of the Laplace approximation (Immer et al., 2021; Cohen et al., 2022; Daxberger et al., 2023). These approaches avoid explicit parameter-space covariance computations and enable more flexible kernel choices, demonstrating strong results in uncertainty estimation and calibration. However, to the best of our knowledge, no prior work has applied the Gaussian process view of Laplace approximations to LoRA fine-tuned transformer models. This setting is particularly well-suited to GP-based inference, since fine-tuning is often performed on relatively small datasets where sparse or variational GP approximations are not required. Moreover, applying Laplace in the GP view scales linearly with the number of parameters, making it especially suitable for large models trained on modest-sized datasets.

We introduce **NTK-LoRA**, a calibration method for LoRA-finetuned Vision Transformers that is simple to implement and requires no specialized knowledge of Gaussian processes, making it accessible to a broad range of researchers and practitioners. Although we focus on binary classification with Vision Transformers, the approach is general and is not limited to vision. NTK-LoRA consistently improves calibration and log-likelihood while preserving or improving predictive accuracy. In this work, we make the following contributions:

- We propose **NTK-LoRA**, the first method that applies the Gaussian process view of the Laplace approximation to *LoRA-finetuned Transformer models*.

- We provide a **simple and practical implementation** that requires no expertise in Bayesian inference or Gaussian processes, making it accessible to practitioners and researchers from other fields.

- We conduct extensive **empirical validation on vision datasets** (CelebA, CUB-200), demonstrating improvements in calibration and uncertainty estimation compared to baselines for Vision Transformers.

- We show that NTK-LoRA can in some cases **perform comparably with fully-trained models after as few as one epoch of fine-tuning**, suggesting potential speedups in low-resource or time-constrained settings.

An overview of the method is provided in Algorithm 1. Code to reproduce our experiments will be made publicly available upon acceptance.

---

[1]Existing Laplace approximation libraries Daxberger et al. (2021) are straightforward to use for small networks, but applying them to large Transformer models can be more difficult in practice, for example due to external dependencies, compatibility with quantized models, or version requirements.

## 2 BACKGROUND

In this section, we review the background needed to understand the theory and motivation behind our method and its relation to the Laplace approximation. Readers primarily interested in implementation details may skip directly to Section 3.3.

### 2.1 LOW-RANK ADAPTATION (LoRA)

LoRA (Hu et al., 2021) is a widely used parameter-efficient fine-tuning method for Transformer models. Like other parameter-efficient approaches, it keeps the pretrained weights frozen while introducing a small number of additional trainable parameters. Concretely, before fine-tuning, LoRA inserts low-rank adapters into selected attention layers by replacing each projection matrix $W$ with $W + AB$, where $A$ and $B$ define a low-rank decomposition whose parameter count is much smaller than that of $W$. This approach offers several benefits, including lower memory usage during training, reduced catastrophic forgetting, and the ability to store many fine-tuned models efficiently.

### 2.2 BAYESIAN INFERENCE

The Bayesian approach treats model parameters probabilistically. Instead of a single vector of parameter values, it considers their joint probability distribution. A prior distribution $p(w)$ is specified and, after observing data, updated into a posterior $p(w|\mathcal{D})$ according to Bayes' theorem:

$$p(w|\mathcal{D}) = \frac{p(\mathcal{D}|w)p(w)}{p(\mathcal{D})} \propto p(\mathcal{D}|w)p(w),$$

where $p(\mathcal{D}|w)$ denotes the likelihood of the observed data $\mathcal{D}$ given model weights $w$. To make predictions for a new, unseen test input $x_*$, the predictive distribution is computed as

$$p(y_*|x_*) = \int p(y_*|x_*, w)p(w|\mathcal{D})dw.$$

This predictive distribution assigns probabilities to all possible outputs, providing more information than a single point prediction. For instance, concentrated probability mass indicates low uncertainty, while a more diffuse distribution reflects higher uncertainty. In situations where only a point estimate is required, the most probable output can be selected as $y_{\text{MAP}} = \arg\max p(y_* \mid x_*)$, known as Maximum a Posteriori (MAP) estimation. Bayesian methods offer key advantages: they capture uncertainty in predictions and allow prior knowledge to be incorporated through the prior distribution. However, for large models the exact predictive distribution is intractable and requires approximate inference.

### 2.3 LINEARIZED LAPLACE

Bayesian inference can be applied even to models with a large number of parameters, such as neural networks, when suitable approximations are used. One such method is the Linearized Laplace (LLaplace) approximation (Daxberger et al., 2021), which combines two approximations to yield a closed-form predictive distribution. First, the posterior is locally approximated by a Gaussian,

$$p(w|\mathcal{D}) \approx q(w) = \mathcal{N}(\mu_w, \Sigma_w),$$

where $\mu_w = w^*$ are the trained model weights and $\Sigma_w$ is the weight covariance, which can be approximated using the Jacobian of the output logits with respect to the weights. For the second approximation, a fixed test input $x_*$ is assumed, and the neural network is viewed as a function of its weights, $f(w) = f(x_*; w)$. The function $f(w)$ can then be approximated in weight space by a first-order Taylor expansion around the trained weights $w^*$:

$$f(x; w) \approx f_{\text{lin}}(x; w) = f(x; w^*) + J_{w^*}(x)(w - w^*),$$

where $J_{w^*} = \nabla_w f(w)\big|_{w=w^*}$ is the Jacobian of the output logits with respect to the weights $w$, and $w^*$ denotes the trained model parameters.

## 2.4 LINEARIZED LAPLACE AS BAYESIAN LINEAR REGRESSION

LLaplace can be interpreted as approximating the neural network with a Bayesian linear regression model in weight space, where the basis functions correspond to Jacobians of the network outputs with respect to the weights. Let $\phi(x) = J_{w^*}(x)^\top$ denote the transposed Jacobian features, and define the deterministic mean function as $b(x) = f(x; w^*) - J_{w^*}(x)w^*$. With output noise $\epsilon$, the LLaplace model can then be expressed as

$$
\begin{aligned}
y &= f_{lin}(x; w) + \epsilon \\
&= \phi(x)^{\mathrm{T}}w + b(x) + \epsilon \\
&= \phi'(x)^{\mathrm{T}}w' + \epsilon
\end{aligned}
$$

The mean function $b(x)$ can be eliminated by transforming the dataset as $\tilde{y} = \phi(x)^\top w + \epsilon$. Alternatively, the basis function can be augmented to include the bias term, $\phi'(x) = (\phi(x), b(x))$, with a corresponding extended weight vector $w' = (w, 1)$. For simplicity of notation, we will write $y = \phi(x)^\top w + \epsilon$.

Bayesian linear regression with Gaussian weights and output noise yields a closed-form predictive distribution $\mathcal{N}(\mu_y, \Sigma_y)$. However, when the number of parameters is large (e.g., $10^4$ or more), the full covariance matrix becomes prohibitively large to compute and store. In such cases, approximations are employed to make LLaplace tractable, though often at the cost of reduced performance. For a more detailed introduction to the Laplace approximation, we refer readers to (Daxberger et al., 2021).

## 2.5 LINEARIZED LAPLACE AS GAUSSIAN PROCESS

A Gaussian process (GP) is a collection of random variables such that any finite subset follows a joint Gaussian distribution. A GP is fully specified by its mean function $\mu(x)$ and covariance function (kernel) $k(x, x')$. Equivalently, a GP can be viewed as a distribution over functions $f : X \to Y$, with the property that evaluations at any finite set of inputs yield samples from a multivariate Gaussian distribution.

$$
f(\mathbf{x}) \sim \mathcal{GP}\left(m(\mathbf{x}),\ k(\mathbf{x}, \mathbf{x}')\right)
$$

Mercer's theorem implies that every GP can be expressed as a Bayesian linear regression in some feature space, which may be infinite-dimensional and difficult to find. Conversely, given a Bayesian linear regression model, the corresponding GP representation is straightforward to obtain. For LLaplace with a weight prior $w \sim \mathcal{N}(0, \sigma_w^2 I)$, an equivalent GP with the same predictive distribution can be constructed. This GP has the following kernel.

$$
k(x, x') = \sigma_w^2 J_{w*}(x) J_{w*}(x')^T
$$

This kernel is known as the (scaled) Neural Tangent Kernel (NTK), which plays an important role in the theory of infinitely wide neural networks Lee et al. (2018); Garriga-Alonso et al. (2019); Novak et al. (2019). In our context, the key property is that a GP with this kernel yields the same predictive distribution as the Linearized Laplace approximation. This perspective is often referred to as GP-Laplace or Function-Space Laplace (Immer et al., 2021; Daxberger et al., 2023).

## 3 METHOD

Our method applies post-hoc calibration to a fine-tuned Transformer model using Low-Rank Adaptation (LoRA) (Hu et al., 2021). We focus on binary classification, though with minor modifications the approach could also be extended to multi-class problems and regression. To perform calibration, only the parameters updated during fine-tuning (i.e., the LoRA weights and the final classification layer) are treated as probabilistic and denoted by $w$, while the remaining pretrained weights are kept

fixed. We apply the Laplace approximation to the selected weights $w$ in function space, as a Gaussian process. Compared to parameter-space Laplace, this approach offers simpler implementation and greater flexibility for incorporating priors.

## 3.1 GP KERNEL

Our GP kernel is a simple linear kernel defined on feature vectors given by transformed Jacobians. Specifically, an input $x$ (e.g., an image or text) is mapped to the Jacobian of the two output logits with respect to the weights $w$, evaluated at the trained parameters $w^*$:

$$ J(x) = \nabla_w f(x; w)|_{w=w^*} \quad f(x; w) \in \mathbb{R}^2. $$

The Jacobians are reshaped from $2 \times d$ matrixes into feature vectors $\tilde{\phi}(x) = vec(J_{w^*}(x))$ of length $2d$, where $d$ is the number of parameters. The feature vectors are then normalized as $\phi(x) = (\tilde{\phi}(x) - \mu_\phi)/\sigma_\phi$, where $\mu_\phi$ and $\sigma_\phi$ are estimated on the training data. This leads to a better numerical stability and gives individual weights more balanced contribution. From Bayesian perspective, we can interpret this as having a diagonal covariance prior that corresponds to the scaling values. The final kernel is as a dot product between two feature vectors with an optional noise kernel for the output noise.

$$ k(x, x') = \sigma_{prior}^2 \phi(x)^T \phi(x') + \sigma_{noise}^2 \delta_{xx'}, $$

where $\delta_{xx'}$ is Kronecker delta, $\sigma_{prior}^2$ is prior variance for weights and $\sigma_{noise}^2$ is prior variance for the output noise — two hyper-parameters fitted on the training data. Our kernel is a scalar kernel — i.e. it outputs a single scalar number instead of $c \times c$ covariance between $c$ outputs of two samples $x$ and $x'$. Outputting a single logit is sufficient for binary classification, improves computational efficiency, and simplifies the implementation.

## 3.2 BINARY CLASSIFICATION

To fit the GP model and make predictions, we use the `GaussianProcessClassifier` (GPC) from scikit-learn (version 1.7). Scikit-learn is a widely used ML library that simplifies implementation and ensures reproducibility. GPC transforms the output logits $z$ into probabilities using the logistic link function (sigmoid), $P(y = 1 \mid z) = (1 + e^{-z})^{-1}$, which gives the probability of the positive class conditioned on logit. In GPC, logits are treated as latent variables since binary labels cannot be represented directly in logit space. Fitting the GP estimates the kernel hyperparameters (prior and noise variance) by maximizing the marginal log-likelihood.

## 3.3 IMPLEMENTATION

Implementing our method requires only 20–30 lines of code and two standard ML libraries: scikit-learn and PyTorch (or any library with automatic differentiation). This compactness allows us to include the full implementation within this paper. Jacobians are computed using `torch.autograd.grad`; for simplicity, we iterate over individual data samples, though a batched implementation could improve performance.

After computing the Jacobians for all training samples and reshaping them into feature vectors, we normalize the features using `StandardScaler` from sklearn. We then fit a GPC, which automatically determines the optimal prior variance (scale of the linear kernel) and noise variance. Our kernel can be easily extended with non-linear kernels or more complex priors, implemented as transformations of the feature vectors. For prediction, we compute Jacobians for the test samples, transform them into feature vectors, and use `predict_proba` from scikit-learn to obtain calibrated probabilities. Optionally, `latent_mean_and_variance` (available in scikit-learn version 1.7 or later) can be used to obtain a distribution over logits, which allows estimating second-order uncertainty.

Our implementation is not restricted to LoRA parameters. Any other parameter-efficient fine-tuning method, or any chosen subset of weights, could be used in place of the LoRA parameters. For regression tasks, `GaussianProcessClassifier` can be replaced with `GaussianProcessRegressor`. In the multi-class setting, the same implementation can be

```python
def fit_ntk_lora(j_train, y_train):
    scale = ConstantKernel(
        constant_value=1.0,
        constant_value_bounds=(1e-5, 1e5))
    noise_kernel = WhiteKernel(
        noise_level=1e-3,
        noise_level_bounds=[1e-7, 1e5])

    kernel = scale * DotProduct() + noise_kernel
    gpc = GaussianProcessClassifier(kernel=kernel)
    scaler = preprocessing.StandardScaler()

    j_train_scaled = scaler.fit_transform(j_train)
    gpc.fit(j_train_scaled, y_train)
    return gpc, scaler
```

```python
def predict_with_ntk_lora(
        j_test, gpc, scaler,
        return_logit_variance=False):
    j_test_scaled = scaler.transform(j_test)
    pred = gpc.predict_proba(j_test_scaled)

    # Optionally, return the whole predictive
    # distribution on logits
    if return_logit_variance:
        z_mean, z_var = \
            gpc.latent_mean_and_variance(
                j_test_scaled)
        return pred, z_mean, z_var

    return pred
```

Figure 1: **Calibration with NTK-LoRA.** Left: we take Jacobians as feature vectors, normalize them, and fit a Gaussian Process Classifier with the kernel defined at Section 3.1. Right: prediction code that returns calibrated probabilities, optionally, full Gaussian predictive distribution over logits is also returned for uncertainty estimation.

applied, although results may be suboptimal, since GPC does not natively support multi-class classification and instead trains multiple one-vs-rest binary classifiers. For the complete implementation, see Figure 1 and Figure 3 (Appendix, second figure).

## 4 EXPERIMENTS

### 4.1 EXPERIMENTAL SETUP

We evaluate our method across six diverse binary classification tasks spanning two different vision domains: CelebA - a large-scale dataset of celebrity face images with binary attribute annotations commonly used to evaluate models for calibration, and CUB-200 – a fine-grained bird classification dataset with binary attribute labels.

From each dataset, we evaluate on three challenging attributes. For CelebA, we focus on *blurry*, *pale skin*, and *rosy cheeks*, which have been noted as inconsistent or noisy (Wu et al., 2023; Lingenfelter et al., 2022); we omit attributes where fine-tuning provides no improvement over the pretrained model. To the best of our knowledge, no prior work systematically identifies the most difficult attributes in CUB. We therefore used a large language model to suggest the most difficult attributes for our experiments: *buff underparts*, *olive back*, and *bill length shorter than head*.

Since calibration is particularly important for models trained on small datasets, we randomly sample 1,000 training images with balanced positive and negative classes. An equally sized test set is constructed in the same way, except for the second CUB attribute, which contains only about 350 training and 350 test samples after balanced sampling. We fine-tune DeiT-Tiny, the smallest variant of the Data-efficient Image Transformer (DeiT) (Touvron et al., 2021), pre-trained on ImageNet-1k (1M images across 1,000 classes). The full set of training hyperparameters is provided in Appendix B.

### 4.2 MAIN RESULTS

In our main experiments, we apply early stopping and select the model with the best validation loss. We compare our method against temperature scaling (Guo et al., 2017) and Laplace with Kronecker-factored covariance, implemented using the Laplace library (Daxberger et al., 2021). Each experiment is repeated 5–8 times with different random seeds, and the averaged results are reported in Table 4.2. We evaluate three main metrics: accuracy (ACC), negative log-likelihood (NLL), and expected calibration error (ECE). In addition, we report accuracy on the top $k\%$ most certain predictions ($k \in \{10, 20, 30\}$), where uncertainty is estimated using the Bernoulli variance.

NTK-LoRA achieves performance comparable to or better than temperature scaling (TS), and consistently outperforms both Laplace and the uncalibrated baseline. We attribute the weaker per-

Table 1: **Results on six binary classification tasks across CelebA and CUB.** CelebA attributes: *blurry (1)*, *pale skin (2)*, *rosy cheeks (3)*; CUB attributes: *buff underparts (1)*, *olive back (2)*, *bill length shorter than head (3)*. All models are calibrated using the checkpoint with the best validation loss. Compared methods: (1) baseline (no calibration), (2) temperature scaling, (3) Laplace with Kronecker-factored covariance, and (4) NTK-LoRA (ours). Best and near-best results are shown in bold.

| CUB • Attr1 | ACC ↑ | NLL ↓ | ECE ↓ | ACC 10%↑ | ACC 20%↑ | ACC 30%↑ |
|---|---|---|---|---|---|---|
| Baseline | 0.7103 | 0.5976 | 0.1028 | 0.8583 | 0.8542 | 0.8339 |
| Temperature | 0.7103 | **0.5678** | 0.0528 | 0.8583 | 0.8542 | 0.8339 |
| Laplace | 0.7070 | 0.6056 | 0.1094 | 0.8550 | 0.8508 | 0.8322 |
| NTK-LoRA (ours) | 0.7150 | **0.5665** | **0.0433** | **0.8817** | 0.8575 | **0.8494** |

| CUB • Attr2 | ACC ↑ | NLL ↓ | ECE ↓ | ACC 10%↑ | ACC 20%↑ | ACC 30%↑ |
|---|---|---|---|---|---|---|
| Baseline | 0.7723 | 0.5510 | 0.1535 | 0.9188 | 0.9203 | 0.9128 |
| Temperature | 0.7723 | **0.4858** | 0.0553 | 0.9188 | 0.9203 | 0.9128 |
| Laplace | 0.7723 | 0.5645 | 0.1674 | 0.9062 | 0.9188 | 0.9097 |
| NTK-LoRA (ours) | **0.7950** | **0.4645** | 0.0817 | **0.9438** | **0.9359** | **0.9244** |

| CUB • Attr3 | ACC ↑ | NLL ↓ | ECE ↓ | ACC 10%↑ | ACC 20%↑ | ACC 30%↑ |
|---|---|---|---|---|---|---|
| Baseline | 0.7208 | 0.5911 | 0.1004 | 0.9020 | 0.8790 | 0.8527 |
| Temperature | 0.7208 | **0.5636** | 0.0525 | 0.9020 | 0.8790 | 0.8527 |
| Laplace | 0.7208 | 0.5961 | 0.1083 | 0.9000 | 0.8760 | 0.8520 |
| NTK-LoRA (ours) | 0.7266 | **0.5620** | **0.0456** | 0.9040 | 0.8780 | **0.8600** |

| CelebA • Attr1 | ACC ↑ | NLL ↓ | ECE ↓ | ACC 10%↑ | ACC 20%↑ | ACC 30%↑ |
|---|---|---|---|---|---|---|
| Baseline | 0.8333 | 0.4899 | 0.1696 | 0.9912 | 0.9844 | 0.9750 |
| Temperature | 0.8333 | **0.3930** | 0.0542 | 0.9912 | 0.9844 | 0.9750 |
| Laplace | 0.8334 | 0.5081 | 0.1916 | 0.9912 | **0.9862** | 0.9754 |
| NTK-LoRA (ours) | 0.8346 | **0.3769** | 0.0421 | 0.9912 | 0.9838 | **0.9767** |

| CelebA • Attr2 | ACC ↑ | NLL ↓ | ECE ↓ | ACC 10%↑ | ACC 20%↑ | ACC 30%↑ |
|---|---|---|---|---|---|---|
| Baseline | 0.7865 | 0.5308 | 0.1519 | 0.9625 | 0.9500 | 0.9296 |
| Temperature | 0.7865 | **0.4600** | 0.0513 | 0.9625 | 0.9500 | 0.9296 |
| Laplace | 0.7865 | 0.5456 | 0.1707 | 0.9638 | 0.9475 | 0.9296 |
| NTK-LoRA (ours) | 0.7866 | **0.4559** | 0.0453 | **0.9700** | **0.9569** | **0.9354** |

| CelebA • Attr3 | ACC ↑ | NLL ↓ | ECE ↓ | ACC 10%↑ | ACC 20%↑ | ACC 30%↑ |
|---|---|---|---|---|---|---|
| Baseline | 0.8001 | 0.5185 | 0.1508 | 0.9725 | 0.9606 | 0.9421 |
| Temperature | 0.8001 | **0.4460** | 0.0541 | 0.9725 | 0.9606 | 0.9421 |
| Laplace | 0.8001 | 0.5311 | 0.1661 | 0.9713 | 0.9588 | 0.9429 |
| NTK-LoRA (ours) | 0.8044 | **0.4404** | 0.0483 | 0.9488 | 0.9469 | 0.9379 |

formance of Laplace to its use of a Kronecker-factored covariance, which imposes independence assumptions across layers. In contrast, our method does not rely on such assumptions.

## 4.3 CALIBRATION DURING TRAINING

To study the effect of model selection, we calibrate models after each training epoch and report averages over multiple runs in Figure 2. On the CelebA dataset, NTK-LoRA yields more stable performance and is less sensitive to the choice of epoch compared to TS. This property is particularly useful when the validation set is unavailable or too small. An additional advantage is that, even after only a few epochs, NTK-LoRA calibrated models often achieve performance comparable to the fully trained model. On the other hand, our method's performance typically degrades once the model begins to overfit, sometimes even before the uncalibrated baseline reaches its best validation loss. Thus, early stopping is recommended to obtain the best results with our approach.

## 4.4 CALIBRATION WITH LIMITED TRAINING TIME

Using the same setup as in the previous experiments, we evaluate models calibrated after a single training epoch. Results are reported in Table 2. In most cases, our method substantially improves accuracy over the baseline and other calibration approaches. Remarkably, in some settings NTK-LoRA achieves performance close to a fully trained model, even when applied after only a single epoch of fine-tuning.

These results are consistent with prior observations that Gaussian process views of neural networks, such as Neural Tangent Kernels, can provide competitive performance without full training (Lee et al., 2018; Novak et al., 2019), and with results showing that Bayesian posteriors can compensate for limited training by integrating uncertainty (Daxberger et al., 2021). We believe this could point toward a practical strategy for scenarios such as adaptation of vision models on small datasets with strict compute or time budgets (e.q. training on edge-devices), where training only for a short period followed by calibration suffices to obtain models that are both accurate and well-calibrated.

## 5 CONCLUSION

In this work, we introduced NTK-LoRA, a simple post-hoc calibration method for Transformer models fine-tuned for binary classification. Our approach leverages the Gaussian process view of neural

Table 2: **Calibration after one epoch.** Same setting as Table 4.2, except models are trained for a single epoch. NTK-LoRA substantially improves accuracy over other methods, in some cases achieving performance close to the fully trained model (e.q., CelebA attribute 3 – *rosy cheeks*).

| CUB • Attr1 | | | | | |
| --- | --- | --- | --- | --- | --- |
| ACC ↑ | NLL ↓ | ECE ↓ | ACC 10%↑ | ACC 20%↑ | ACC 30%↑ |
| Baseline 0.7042 | 0.6137 | 0.1058 | 0.8400 | 0.8367 | 0.8189 |
| Temperature 0.7042 | **0.5860** | 0.0558 | 0.8400 | 0.8367 | 0.8189 |
| Laplace 0.7042 | 0.6175 | 0.1111 | 0.8417 | 0.8350 | 0.8178 |
| NTK-LoRA (ours) 0.7158 | 0.5676 | **0.0431** | **0.8833** | **0.8558** | **0.8450** |

| CUB • Attr2 | | | | | |
| --- | --- | --- | --- | --- | --- |
| ACC ↑ | NLL ↓ | ECE ↓ | ACC 10%↑ | ACC 20%↑ | ACC 30%↑ |
| Baseline 0.7427 | 0.6111 | 0.1697 | 0.8781 | 0.8547 | 0.8298 |
| Temperature 0.7427 | 0.5516 | **0.0924** | 0.8781 | 0.8547 | 0.8298 |
| Laplace 0.7427 | 0.6182 | 0.1780 | 0.8625 | 0.8578 | 0.8277 |
| NTK-LoRA (ours) **0.8036** | **0.4802** | 0.0914 | **0.9219** | **0.9156** | **0.9055** |

| CUB • Attr3 | | | | | |
| --- | --- | --- | --- | --- | --- |
| ACC ↑ | NLL ↓ | ECE ↓ | ACC 10%↑ | ACC 20%↑ | ACC 30%↑ |
| Baseline 0.7080 | 0.6001 | 0.1018 | 0.8920 | 0.8540 | 0.8313 |
| Temperature 0.7080 | **0.5707** | 0.0475 | 0.8920 | 0.8540 | 0.8313 |
| Laplace 0.7080 | 0.6039 | 0.1096 | 0.8920 | 0.8550 | 0.8287 |
| NTK-LoRA (ours) **0.7232** | 0.5589 | **0.0449** | 0.8960 | **0.8850** | **0.8520** |

| CelebA • Attr1 | | | | | |
| --- | --- | --- | --- | --- | --- |
| ACC ↑ | NLL ↓ | ECE ↓ | ACC 10%↑ | ACC 20%↑ | ACC 30%↑ |
| Baseline 0.7321 | 0.6059 | 0.1554 | 0.9425 | 0.9138 | 0.8912 |
| Temperature 0.7321 | 0.5366 | 0.0627 | 0.9425 | 0.9138 | 0.8912 |
| Laplace 0.7321 | 0.6118 | 0.1627 | 0.9450 | 0.9150 | 0.8925 |
| NTK-LoRA (ours) **0.8099** | **0.4230** | **0.0571** | **0.9925** | **0.9819** | **0.9671** |

| CelebA • Attr2 | | | | | |
| --- | --- | --- | --- | --- | --- |
| ACC ↑ | NLL ↓ | ECE ↓ | ACC 10%↑ | ACC 20%↑ | ACC 30%↑ |
| Baseline 0.6699 | 0.6371 | 0.1097 | 0.8762 | 0.8406 | 0.8121 |
| Temperature 0.6699 | 0.6036 | **0.0540** | 0.8762 | 0.8406 | 0.8121 |
| Laplace 0.6699 | 0.6406 | 0.1152 | 0.8762 | 0.8394 | 0.8146 |
| NTK-LoRA (ours) **0.7492** | **0.5246** | 0.0631 | **0.9438** | **0.9244** | **0.9004** |

| CelebA • Attr3 | | | | | |
| --- | --- | --- | --- | --- | --- |
| ACC ↑ | NLL ↓ | ECE ↓ | ACC 10%↑ | ACC 20%↑ | ACC 30%↑ |
| Baseline 0.7317 | 0.5906 | 0.1251 | 0.9563 | 0.9194 | 0.8938 |
| Temperature 0.7317 | 0.5403 | **0.0583** | 0.9563 | 0.9194 | 0.8938 |
| Laplace 0.7317 | 0.5960 | 0.1341 | 0.9588 | 0.9219 | 0.8933 |
| NTK-LoRA (ours) **0.8004** | **0.4532** | 0.0636 | **0.9662** | **0.9450** | **0.9325** |

networks and can be implemented with only a few lines of code using standard libraries such as PyTorch and scikit-learn, making it accessible to a broad range of practitioners. Across experiments with Vision Transformers on CelebA and CUB, NTK-LoRA achieved performance comparable to or better than temperature scaling in terms of acuracy, negative log-likelihood, and calibration, and generally outperformed Laplace with Kronecker-factored covariance, which often failed to improve over the uncalibrated baseline. Notably, we find that NTK-LoRA can achieve strong performance after only a single epoch of fine-tuning, sometimes approaching the accuracy of fully trained models. This highlights its potential in scenarios where training time or computational resources are limited. Overall, NTK-LoRA provides a practical calibration tool that integrates seamlessly with existing fine-tuning workflows.

# 6 Limitations and Future Work

While our experiments focused on Vision Transformers for binary classification, the approach is applicable to other Transformer-based models (e.g., large language models) fine-tuned with parameter-efficient methods, and with minor modifications could also extend to regression and other tasks. Further experiments are needed to validate its effectiveness in these broader settings. However, since our method is primarily designed for problems with a small number of outputs, scaling it to high-dimensional generative settings (e.g., next-token prediction or image generation) would require developing more efficient extensions that would scale sublinearly in the number of outputs.

Our method performs well for fine-tuned Vision Transformers with a relatively small number of calibrated parameters and training examples (up to a few thousand). There is no inherent restriction on the size of the pretrained model. When calibrating larger parameter sets, the method still scales linearly; however, in preliminary experiments we observed somewhat reduced efficiency in high-dimensional spaces, requiring more careful selection of training hyperparameters and priors, consistent with findings by Cinquin et al. (2021). To address this limitation, future work could explore more suitable priors or employ parameter-efficient fine-tuning methods that adapt fewer parameters than LoRA.

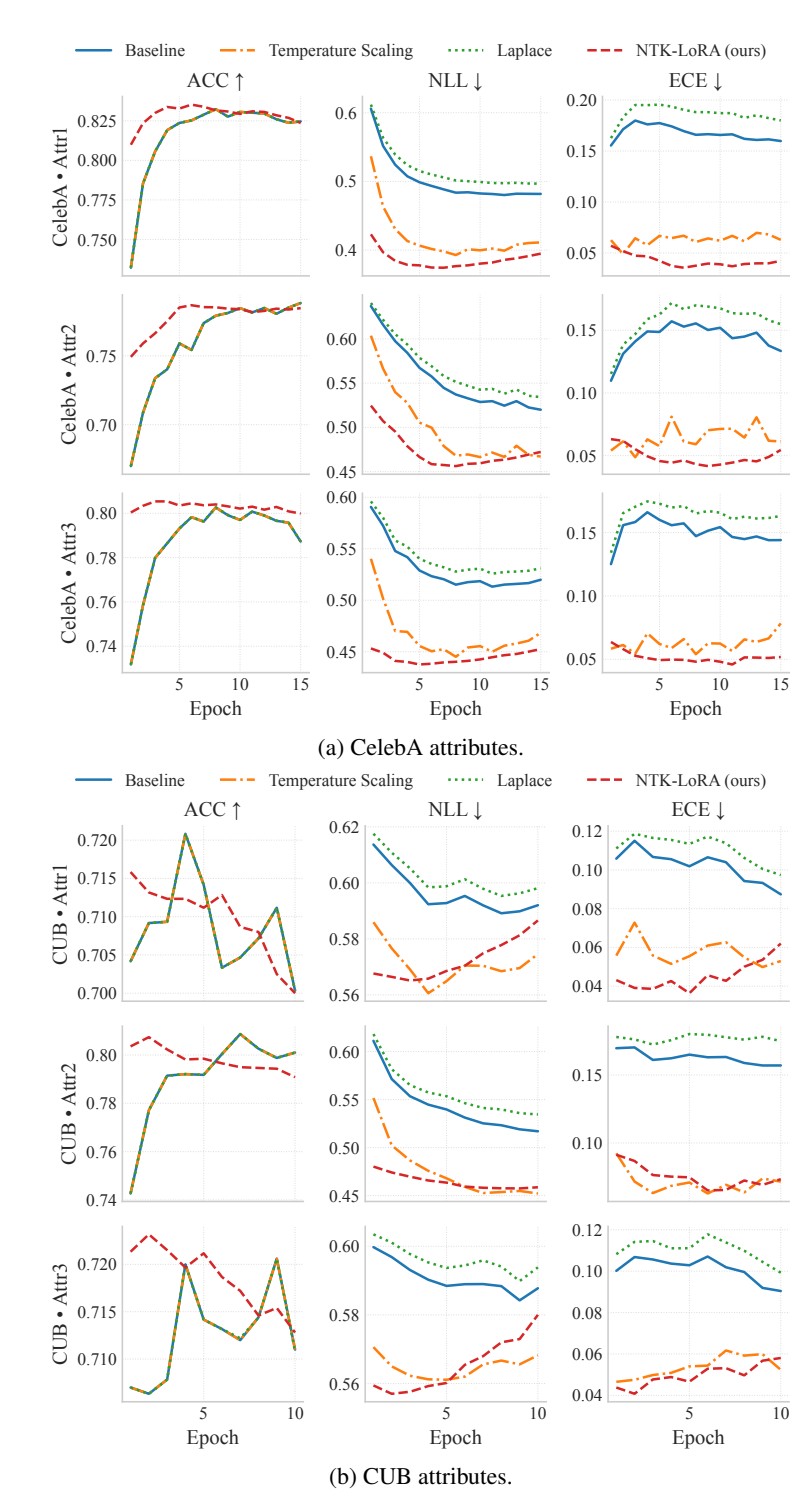

Figure 2: **Calibration during training.** (a) CelebA attributes: *blurry (1)*, *pale skin (2)*, *rosy cheeks (3)*; (b) CUB attributes: *buff underparts (1)*, *olive back (2)*, *bill length shorter than head (3)*. Models are calibrated after each epoch, with results averaged over multiple random seeds. NTK-LoRA provides more stable performance across epochs than TS and outperforms other methods in early epochs.

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

## A  GAUSSIAN PROCESSES

### A.1  DEFINITION AND BASIC PROPERTIES

A Gaussian Process (GP) is a collection of random variables, where finite subsets have a joint Gaussian distribution. Every GP is uniquely specified by its mean $\mu(x)$ and covariance function (kernel) $k(x_1, x_2)$. We can also see it as a distribution on functions $f : X \to Y$, we write:

$$f(\mathbf{x}) \sim \mathcal{GP}\left(m(\mathbf{x}),\ k(\mathbf{x}, \mathbf{x}')\right)$$

Perhaps, a more intuitive way to think about GPs for ML practitioners is as the following Bayesian linear regression model.

$$y = f(x) + \epsilon = w^T \psi(x) + \epsilon$$

, where $\psi(x)$ is a basis function, $w \sim \mathcal{N}(0, \sigma^2 I)$ are the weight and $\epsilon \sim \mathcal{N}(\mu, \Sigma)$ is the output noise. Mercer's theorem tells us every GP is equivalent to a BLR in certain feature space (that could be infinite and difficult to find). On the other hand, going from BLR to GP is straightforward. Given a BLR model, we can define a GP as follows.

$$m(x) = E[f(x)] = \psi(x)^{\mathrm{T}} \mu_w$$
$$k(x_1, x_2) = Cov(f(x_1), f(x_2)) = \psi(x)^T \Sigma_w \psi(x)$$

where $f(x) = w^T \psi(x)$ and $w \sim N(\mu_w, \Sigma_w)$.

This simplifies for $w \sim \mathcal{N}(0, \sigma_{prior}^2 I)$, where $\sigma_{prior}^2$ is the prior variance and $\epsilon \sim \mathcal{N}(0, \sigma_{noise}^2 I)$, where $\sigma_{noise}^2$ is the noise variance, this simplifies to.

$$m(x) = 0 \quad \text{and} \quad k(x_1, x_2) = \sigma_{prior}^2 \psi(x)^T \psi(x)$$

## A.2 COMPUTING PREDICTIONS

To make predictions with GPs, we look at the join distribution of test and training data, which is by the definition Gaussian. From that we can derive the following formula for the predictive distribution that is a Gaussian $N(\mu_{y_*}, \sigma_{y_*}^2)$ for a test input $x_*$ and output $y_*$.

$$\mu_{y_*} = \mathbb{E}\big[f(\mathbf{x}_*) \mid X, \mathbf{y}\big] = \mathbf{k}_*^\top \big(K + \sigma_n^2 I\big)^{-1} \mathbf{y}$$
$$\sigma_*^2 = \text{Var}\big[f(\mathbf{x}_*) \mid X, \mathbf{y}\big] = k(\mathbf{x}_*, \mathbf{x}_*) - \mathbf{k}_*^\top \big(K + \sigma_n^2 I\big)^{-1} \mathbf{k}_*,$$

where

- $K_{ij} = k(\mathbf{x}_i, \mathbf{x}_j)$
- $k_* = \big[k(\mathbf{x}_1, \mathbf{x}_*), \ldots, k(\mathbf{x}_n, \mathbf{x}_*)\big]^\top$,
- $\sigma_n^2$ is the variance of the i.i.d. Gaussian observation noise.

GPs are non-parametric, they don't have any weight vector to fit, only hyper-parameters that can be optimized after observing the data. This is done by maximization of the marginal log-likelihood.

## A.3 BINARY CLASSIFICATION

Classification with Gaussian processes is slightly more involved than regression, since the likelihood is no longer Gaussian. In our experiments we use the `GaussianProcessClassifier` (GPC) implementation from `scikit-learn` (version 1.7). The GPC supports only binary classification directly; multi-class problems are handled internally using a one-vs-rest reduction to multiple binary classifiers. Unlike regression, where the GP represents the predicted values directly, here it represents logits $f$ that have to be converted to class probabilities via a logistic link:

$$P(y = 1 \mid f) = \sigma(f) = \frac{1}{1 + e^{-f}}.$$

Instead of transforming the training labels into logits directly (which would give us infinite values), we treat logits as latent values and find the hyper-parameters (in our case prior and noise variance) by maximization of the marginal log-likelihood.

$$\log p(\mathbf{y} \mid X, \theta) = \log \int p(\mathbf{y} \mid \mathbf{f}) \, p(\mathbf{f} \mid X, \theta) \, d\mathbf{f}$$

Because the Bernoulli likelihood breaks conjugacy, the posterior $p(\mathbf{f} \mid X, y)$ is intractable. By default scikit-learn's `GaussianProcessClassifier` uses the Laplace approximation (similar to the one we use to approximate posterior of the neural network) to form a Gaussian approximation around the MAP latent $\hat{\mathbf{f}}$, and then computes predictive probabilities via one-dimensional quadrature.

## B EXPERIMENTAL SETUP

We use pretrained Vision Transformers from HuggingFace: `DeiT-Tiny` for our main experiments, `DeiT-Small`, and `ViT-Base` (`ImageNet-21k`) for our additional experiments. LoRA fine-tuning is applied to a subset of attention layers (typically the last one), with rank 1 and scaling parameter $\alpha = 4$. Adapters are inserted into the query and value projection matrices. The last layer and LoRA adapters are trained, other model weights remain frozen.

```python
def compute_grad(output, calibrated_params):
    # gradients for calibrated_params
    grads = torch.autograd.grad(
            output,
            calibrated_params,
            create_graph=True)
    return torch.cat([g.reshape([-1])
                    for g in grads])
```

```python
def compute_jacobians(x_sample, model,
                        calibrated_params):
    out = model(x_sample[None])
    gradients = [compute_grad(
                    out[0, i],
                    calibrated_params)
                for i in range(out.shape[1])]
    return torch.stack(gradients, axis=1)
```

Figure 3: **Computing Jacobians with PyTorch autograd.** Given an input sample, the model, and the parameters to calibrate (LoRA adapters and the final layer), we compute the Jacobian of the output logits with respect to those parameters.

We train with the Adam optimizer (learning rate $1.5 \times 10^{-3}$, weight decay 0.01), using a batch size of 32 for 10 epochs for CUB and 15 epochs for CelebA. We repeat experiments with 5-8 random seeds, different seed sets are used for each attribute to ensure robustness.

Experiments are implemented in PyTorch (v2.7) and scikit-learn (v1.7). We use Hugging-Face Transformers for pretrained ViTs. All experiments were conducted on an NVIDIA A40 GPU with 48GB memory. However, our experiments with smaller models (DeiT-Tiny) can be reproduced on cheaper GPUs, on Apple Silicon devices, or even on CPUs without dedicated accelerators.

## C   LLM USAGE

In preparing this work, we used ChatGPT (OpenAI's GPT-4 and GPT-5) as a writing and editing assistant to improve clarity and readability of text passages, suggest alternative phrasings, and provide LaTeX code snippets for tables, figures, and formatting. In addition, large language models were used to generate Python code for figures and visualizations and to suggest suitable datasets for evaluation.

All scientific contributions, research ideas, model development, experiments, analyses, and conclusions are entirely our own. The LLM was not used for the ideation of the method, the design of the experiments, the data analysis, or the generation of the results, but only to provide feedback on the ideas of the authors. The authors take full responsibility for the final content of the paper.

