# OpenReview forum: "NTK-LoRA: Calibrating Fine-tuned Vision Transformers using Gaussian Processes"
_ICLR.cc/2026/Conference — ICLR 2026 Conference Desk Rejected Submission_

### Official Review · Reviewer_JEH4 · 2025-10-31

**Soundness:** 2
**Presentation:** 3
**Contribution:** 3
**Rating:** 4
**Confidence:** 4

**Summary:**

The authors propose a post-hoc calibration method which fits a GP and performs a function-space Laplace approximation over LoRA parameters. They demonstrate that NTK-LoRA outperforms other post-hoc methods like temperature scaling and Laplace with Kronecker-factored covariance over binary classification tasks from CelebA and CUB.

**Strengths:**

- The paper is well-written, and the motivations are clearly explained. The background and method section are sufficiently detailed, and the presentation is easy to follow.
- The proposed method is conceptually intuitive and well-justified . Furthermore, the method is also simple to implement, and the authors demonstrate its implementation using only a few lines of code, making it easily usable by practitioners.
- NTK-LoRA demonstrates strong empirical performance, outperforming other post-hoc calibration methods such as weight-space Laplace and temperature scaling on the two vision datasets benchmarked.

**Weaknesses:**

- The NTK-LoRA approach has significant limitations for where it can be applied. Because it relies on a GP, the method is computationally feasible only for problems with a moderate number of features and a relatively small number of training examples a small number of training points. Furthermore, the current formulation naively extends only to binary classification. Extending this framework beyond any one of these limitations would require substantial amounts of compute or require other approximations or assumptions that are non-trivial.
- The empirical evidence is limited to vision transformers and only two image datasets. The paper could be stronger if the method was shown to work across more diverse models and settings, or shown to be effective in domain shifts, etc. Furthermore, although the chosen baselines are commonly used, it may also strengthen the paper to include further comparisons with additional calibration methods.
- Nit: The tables are also a bit difficult to read, maybe because of the font size? It renders strangely on my pdf reader.

**Questions:**

- Did you perform any ablations to demonstrate how the performance of NTK-LoRA varies as you change different aspects of the problem, such as number of parameters for LoRA, number of fine-tuned points, etc? It would benefit the paper if there were explicit recommendations for the types of problems that this approach should be applied to compared to baselines like temperature scaling.

---

> ### Author Response · Authors · 2025-11-17
> **Response to Reviewer Comments**
>
> Thank you for the detailed feedback and for highlighting both the strengths and areas for improvement. We appreciate the reviewer’s comments and address each of the concerns below.
>
> Limitation to binary classification: Our current formulation focuses on a relatively specific setting: calibration for binary classification with fine-tuned vision models. However, this setting arises very frequently in practice — for example, disease vs no disease in healthcare, defect vs no defect in manufacturing, valid vs invalid in document workflows, as well as many binary decisions in face recognition, safety, and monitoring applications — where reliable uncertainty estimates are crucial.
>
> Empirical evidence: We chose two established datasets with many diverse binary attributes to allow controlled comparisons. We also ran additional experiments with larger ViTs, but these were not included due to space and time constraints. We agree that expanding to more datasets, models, domain-shift settings, and additional calibration baselines would further strengthen the work, and we will consider adding some of these in a future revision
>
> Formatting of tables: Thank you for pointing this out. We will try to adjust the table formatting and font sizes in the revision to ensure clearer rendering.
>
> Ablations: In preliminary (non-included) experiments, we varied both the LoRA rank and the size of the fine-tuning dataset. We observed that NTK-LoRA performs similarly across these settings as long as the base model does not overfit heavily. Overfitting tended to appear when using LoRA ranks above 2 or when fewer than 500 training examples were available. We agree that more systematic ablations would strengthen the paper, and we will consider including it in future revisions.

---

### Official Review · Reviewer_PWSA · 2025-10-31

**Soundness:** 2
**Presentation:** 2
**Contribution:** 2
**Rating:** 2
**Confidence:** 3

**Summary:**

The paper tackles the problem of miscalibration (overconfidence issue) while finetuning vision transformer models using LoRA based parameter efficient fine tuning. The proposed solution is a post-hoc method that performs Laplace approximation of the posterior distribution. Experiments conducted with binary classification settings demonstrate the effectiveness of the proposed solution. The paper emperically justifies the proposed solution in comparison to linearized Laplace as Gaussian process.

**Strengths:**

- The paper tackles important problem of miscalibration for transformers based fine-tuning for downstream tasks.
- The solution is simple and does not require to tune hyperparameter.

**Weaknesses:**

- The proposed method is limited to binary classification.
- [1] also leverages Laplace approximation for fine-tuning LLMs. In introduction, the paper discusses that implementing [1] in practice is challenging. However, [1] proposes Laplace approximation to a larger scale models than in this paper. How is the proposed method superior in practice than [1]? Can the authors clarify? It is suggested to compare with this method directly.
- The paper does not include a discussion and comparison with recent literature works in calibration that are directly relevant to the proposed work [2,3, 4].
- The motivation is for fine-tuning transformer models (including vision and LLMs), but the experiments are limited to vision transformers and Lora as parameter-efficient fine-tuning (PEFT) methods. The other PEFT methods include, but are not limited to prompt tuning, bias tuning, adapter and vera.
- The experiments are not comprehensive to verify the superiority of the method.

References

[1] Yang, Adam X., et al. "Bayesian Low-rank Adaptation for Large Language Models." The Twelfth International Conference on Learning Representations.

[2] Guo, Chuan, et al. "On calibration of modern neural networks." International conference on machine learning. PMLR, 2017.

[3] Pandey, Deep, Spandan Pyakurel, and Qi Yu. "Be confident in what you know: Bayesian parameter efficient fine-tuning of vision foundation models." Advances in Neural Information Processing Systems 37 (2024): 44814-44844.

[4] Chen, Lin, et al. "Vit-calibrator: Decision stream calibration for vision transformer." Proceedings of the AAAI Conference on Artificial Intelligence. Vol. 38. No. 2. 2024.

**Questions:**

Please refer to weakness section

---

> ### Author Response · Authors · 2025-11-17
> **Response to Reviewer Comments**
>
> Thank you for the thoughtful assessment and constructive suggestions. We address the concerns raised in the weaknesses section below.
>
> Limitation to binary classification: Our experiments focus on binary classification because this is a common and practically important setting for fine-tuned vision models—e.g., medical diagnosis (presence/absence), industrial defect detection, document verification, and various binary detection tasks in security and face analysis—where reliable calibration is crucial. Although we evaluate this specific case, the approach itself is not limited to binary outputs and can also be applied to other settings, such as image-based regression. Extending NTK-LoRA to multi-class and language settings is an important direction for future work.
>
> Regarding Bayesian Lora: We use [1] (Bayesian LoRA) as our Laplace baseline in all experiments. In our setup, we found that weight-space Laplace with Kronecker-factored covariance did not improve over the uncalibrated baseline on binary classification tasks, and we were unable to reproduce the gains reported for LLMs in [1] under our settings.
> The key practical difference is computational: [1] applies Laplace in weight space, where the cost scales cubically with the number of adapted parameters. To remain tractable for large models, [1] requires Kronecker-factored covariances (that assumes independence between layers) and additional implementation complexity.
> Our method performs Laplace in function space, using Jacobian features. This corresponds to weight-space Laplace with a full covariance but avoids explicitly forming or approximating that covariance. As a result, for small–moderate fine-tuning datasets, our approach is simpler to implement and scales more favorably with model size. Moreover, our method is easier to implement and requires only standard libraries (Pytorch and sklearn). On the other hand, our method has cubic complexity in the number of training samples (due to the GP), which is manageable in our setting but would require sparse/variational GP approximations for larger datasets.
>
> Regarding baselines: Thank you for pointing out these related works. We agree that additional baselines would further strengthen the paper. Temperature Scaling [2] is used as our primary calibration baseline and is compared against NTK-LoRA throughout the experiments. Bayesian-PEFT [3] improves calibration in few-shot multi-class settings via evidential learning and ensembles during fine-tuning; however, it targets a different regime than ours, which focuses on calibrating models fine-tuned on moderately sized datasets (~1000 samples). ViT-Calibrator [4], despite its name, is aimed at improving accuracy by modifying internal ViT representations rather than providing calibrated probabilities, so it is not a suitable baseline for our calibration-focused method.
>
> We agree that including additional PEFT methods (e.g., VERA) would further strengthen the evaluation. We focused on LoRA because it is by far the most widely used and strongest PEFT approach in practice. We also ran preliminary experiments with other PEFT methods and observed similar results, but due to time and space constraints these results were not included. We will consider extending the study to additional PEFT variants in future work.

---

### Official Review · Reviewer_fVnv · 2025-11-01

**Soundness:** 2
**Presentation:** 2
**Contribution:** 2
**Rating:** 2
**Confidence:** 3

**Summary:**

Fine-tuning large pretrained Transformer models (e.g., Vision Transformers and LLMs) on small domain-specific datasets often leads to overconfident and poorly calibrated predictions. While post-hoc calibration methods like temperature scaling (TS) are simple, they cannot address complex miscalibrations. This paper proposes NTK-LoRA, a Bayesian calibration method for fine-tuning Transformer models. To perform Bayesian uncertainty quantification, the authors use a Gaussian process view of neural networks and apply a linear Laplace approximation to make posterior computation tractable. The proposed method uses linearized Jacobian features to form the GP Kernel and, as a result, incorporates prior knowledge without requiring any hyperparameter tuning. The paper primarily focuses on solving binary classification tasks, and the given experiments achieve better or comparable performance to baselines.

**Strengths:**

1) The paper proposes a novel connection between LoRA fine-tuning with Gaussian Process–based calibration through the NTK perspective. The proposed method is lightweight, requiring minimal code and standard libraries — practical for broad adoption.

2) The evaluations across the two datasets and multiple attributes demonstrate consistent, although small, performance gains. The finding that NTK-LoRA can match full-finetuning performance after a single epoch is quite interesting. Understanding and replicating this result in more complex settings, such as multi-class classification or generation, would be highly impactful.

**Weaknesses:**

1) The experiments are restricted to binary classification on small datasets with ViTs; claims about generality to LLMs or multi-class settings remain unvalidated. It would be good to have some non-binary classification tests. It is unclear how well the kernel Idea generalizes to multiclass settings.

2) The improvements over Temperature Scaling, though consistent, are numerically small in some cases (ECE drops of ~0.01–0.02). The runtime complexity of needing Jacobian evaluations might not be justified. The method’s reliance on per-sample Jacobian computation could become computationally expensive for larger parameter subsets or multi-class models. It would be good to see some results comparing the

3) While the motivation for using NTK-inspired Kernels is interesting, the explanation feels a bit hand-wavy. From a Bayesian perspective, the authors claim that using the Linear Laplace approximation yields an NTK-style formulation with Jacobian features. This idea is not adequately explained. It would be great to have a formal result showing the sequence of assumptions that yield the final Jacobian feature form. The methodology is not adequately motivated.

**Questions:**

Please refer to the Weaknesses section for the questions.

---

> ### Author Response · Authors · 2025-11-17
> **Response to Reviewer Comments**
>
> We appreciate the reviewer’s careful assessment. Our responses to the points regarding scope, empirical gains, efficiency, and theoretical explanation are provided below.
>
> Regarding binary classification:
> Our experiments focus on binary classification with vision models fine-tuned on small datasets, as this is a common and practically important setting—e.g., medical diagnosis (presence/absence), industrial defect detection, document verification, and security-related binary decisions—where reliable calibration is crucial. Although we evaluate this specific case, the approach itself is not limited to binary outputs and can also be applied to other tasks, such as image-based regression. We did not intend to claim empirical validation on LLMs; rather, we aimed to highlight that the method is model-agnostic when LoRA is used. Extending NTK-LoRA to multi-class and language settings is an important direction for future work.
>
> Modest improvements over TS and Jacobian computation cost:
> We agree that in some settings the improvements over Temperature Scaling are numerically small,
> but NTK-LoRA consistently reduces both NLL and ECE across all evaluated tasks.
> Moreover, in early-epoch fine-tuning, the method yields substantially larger gains
> and often matches near-fully-trained accuracy.
> The Jacobian cost is low in our binary setting (two outputs) and when LoRA is applied only to the last layers.
> We agree that scaling to settings with many output classes (e.g., generative models)
> would require additional considerations, which is part of our ongoing research.
>
> NTK/GP motivation:
> Since the paper aims to provide a practical calibration method for binary classification with PEFT,
> we focused the theoretical discussion on the main result and referred readers to
> prior work for the full derivation.
> We agree that adding a more detailed summary of the underlying assumptions and the resulting
> NTK-style kernel would improve clarity, and we will consider revising this section accordingly.

---

### Official Review · Reviewer_UQgj · 2025-11-03

**Soundness:** 3
**Presentation:** 3
**Contribution:** 2
**Rating:** 4
**Confidence:** 4

**Summary:**

This paper proposes using Gaussian processes with LoRA Jacobian as a feature vector to calibrate uncertainty during fine-tuning.

**Strengths:**

The writing is clean and easy to read. The paper combines GP with PEFT tuning methods, providing a calibration measure for output prediction; such a combination is novel.

**Weaknesses:**

1. In the current setting, the LoRA rank is 1. Larger-scale fine-tuning results are missing (e.g. larger rank). It would be beneficial to show the method works with higher LoRA ranks and generalizes to other PEFT tuning methods.
2. More recent baselines are missing. To incorporate uncertainty for model prediction, Bayesian (last) Layer [1, 3] or inducing uncertainty priors for the last layer during fine-tuning [2] could be less computationally expensive and should be compared with the proposed method.

[1] Variational Bayesian Last Layers

[2] Fine-Tuning with Uncertainty-Aware Priors Makes Vision and Language Foundation Models More Reliable

[3] On Last-Layer Algorithms for Classification: Decoupling Representation from Uncertainty Estimation

**Questions:**

### Questions

- Line 661: What do the 5–8 random seeds mean?
- Figure 2: Is this performance on the test set? Could you provide the best epoch chosen by the validation set to ensure the method is robust to the selection criteria?
- What is the computational cost of NTK-LoRA? Could you provide the time cost? Do we need all layer gradient features to have good uncertainty estimation?

---

> ### Author Response · Authors · 2025-11-17
> **Response to Reviewer Comments**
>
> Thank you for the constructive review and positive comments. We address the concerns about LoRA rank, baselines, and experimental details as follows.
>
> LoRA rank and other PEFT configurations:
> We focused on rank-1 LoRA to prevent overfitting given the relatively small datasets.
> We also ran preliminary experiments with higher ranks and other PEFT methods and observed similar trends,
> but did not include these results due to time and space constraints.
>
>
> Regarding additional baselines:
> Thank you for highlighting these works.
> We agree that incorporating additional baselines such as variational methods would further strengthen the evaluation,
> and we will consider adding such baselines in future revisions.
>
> Concerning the number of random seeds:
> “5--8 random seeds” means that each experiment was repeated 5 to 8 times with different random seeds
> to ensure stable and robust results.
>
> Regarding the evaluation shown in the figures:
> Yes, all results shown in the figures are evaluated on the test set.
> While we could also store the best epochs selected by the validation set,
> producing these plots would require generating new figures.
> The current figures show averages over multiple training runs with possibly different best epochs.
>
> On the computational cost of the method:
> Computing Jacobians for binary classification corresponds to approximately two backward passes
> (one per output dimension).
> When LoRA is applied only to the last $k$ attention blocks,
> the backward pass can be restricted to these layers, in addition to a single forward pass.
> The kernel matrix computation costs $O(n^{2} d)$, where $n$ is the number of training samples
> and $d$ is the number of adapted parameters (i.e., $O(n^{2})$ dot products),
> and fitting the Gaussian process costs $O(n^{3})$.
> In practice, computing calibrated probabilities with our method took roughly one minute for $n \approx 1000$.

---

### Note · Program_Chairs · 2026-01-17
**Submission Desk Rejected by Program Chairs**

The following references in this submission do not refer to real documents and/or have major errors in bibliographic information:

 Jeremy M. Cohen, Erik A. Daxberger, Zachary Nado, James Martens, Mark van der Wilk, and Richard E. Turner. Validating neural network uncertainty using the laplace approximation. In Advances in Neural Information Processing Systems (NeurIPS), 2022